# Alcohol Consumption, Drinking Patterns, and Cognitive Performance in Young Adults: A Cross-Sectional and Longitudinal Analysis

**DOI:** 10.3390/nu12010200

**Published:** 2020-01-13

**Authors:** Henk Hendriks, Ondine van de Rest, Almar Snippe, Jasper Kieboom, Koen Hogenelst

**Affiliations:** 1Hendriks Nutrition Support for Business, 3703BP Zeist, The Netherlands; 2Division of Human Nutrition and Health, Wageningen University & Research, 6708WE Wageningen, The Netherlands; ondine.vanderest@wur.nl; 3TNO (The Netherlands Organization for Applied Scientific Research), 3704HE Zeist, The Netherlands; almar.snippe@tno.nl (A.S.); jasper.kieboom@tno.nl (J.K.); koen.hogenelst@tno.nl (K.H.)

**Keywords:** young adult, alcohol consumption, cognitive performance

## Abstract

Long-term alcohol abuse is associated with poorer cognitive performance. However, the associations between light and moderate drinking and cognitive performance are less clear. We assessed this association via cross-sectional and longitudinal analyses in a sample of 702 Dutch students. At baseline, alcohol consumption was assessed using questionnaires and ecological momentary assessment (EMA) across four weeks (‘Wave 1’). Subsequently, cognitive performance, including memory, planning, and reasoning, was assessed at home using six standard cognition tests presented through an online platform. A year later, 436 students completed the four weeks of EMA and online cognitive testing (‘Wave 2’). In both waves, there was no association between alcohol consumption and cognitive performance. Further, alcohol consumption during Wave 1 was not related to cognitive performance at Wave 2. In addition, EMA-data-based drinking patterns, which varied widely between persons but were relatively consistent over time within persons, were also not associated with cognitive performance. Post-hoc analyses of cognitive performance revealed higher within-person variance scores (from Wave 1 to Wave 2) than between-person variance scores (both Wave 1 and Wave 2). In conclusion, no association was observed between alcohol consumption and cognitive performance in a large Dutch student sample. However, the online cognitive tests performed at home may not have been sensitive enough to pick up differences in cognitive performance associated with alcohol consumption.

## 1. Introduction

Alcohol consumption is common among adolescents and young adults [1]. Particularly at this young age when brains are still maturing [2], both structurally [3] and functionally [4], alcohol consumption may have detrimental long-term neurological effects. However, the extent to which alcohol consumption, both in terms of dosing (e.g., consumption levels) and exposure (e.g., frequency and years of drinking), leads to neural or cognitive problems is unclear.

Excessive alcohol consumption is assumed to be neurotoxic [5], and it is well known that long-term alcohol abuse in adults leads to neurodegeneration. Affected regions in the brain are particularly the prefrontal cortex and the hippocampus [6]. Such neurodegeneration may lead to functional deficits, such as impairment of executive and motivational functions that guide self-regulation and goal-directed behavior [6]. Young adults are extra vulnerable because reward neurocircuitry undergoes significant developmental shifts, making the reward system particularly sensitive to alcohol in ways that could promote excessive consumption [7]. 

Cross-sectional studies in young adult binge drinkers showed impaired executive cognitive performance and brain development compared to nonbinge-drinking control subjects, as suggested by electrophysiological differences during the execution of a visual task with a high working memory load [8,9,10]. Longitudinal observational studies, in which binge-drinking adolescents were followed into adulthood, showed that long-term excessive drinking was associated with negatively affected working memory [11] and verbal memory and learning [12] as opposed to nonbinge-drinking adolescents. Similarly, college students who reported regular binge drinking performed poorer on planning and spatial working memory tasks as compared to nondrinking controls [13]. 

In the general population, only a few small-scale longitudinal studies on alcohol consumption and cognitive performance have been performed. A report on both a cross-sectional study and a three-year longitudinal study in adolescents showed altered fMRI responses indicative of less efficient information processing in 20 heavy drinkers, but not in 20 nondrinkers [14]. Another report on the same group of adolescents showed less activation of inhibitory circuitry before the onset of heavy drinking [15]. After transitioning into heavy drinking, the participants showed more activation during response inhibition than nondrinking controls. This suggests that neural vulnerabilities exist prior to the onset of substance use, and the initiation of heavy drinking may lead to additional alterations in brain functioning [15]. 

Whereas the association between heavy drinking and cognitive performance has been well studied, there are few observational studies that examined the association between light or moderate drinking and cognitive performance in young drinkers [16]. One study assessed cognitive performance in students who were either abstainers or light or moderate drinkers. The study found no differences between drinking groups in terms of cognitive performance; however, the study was cross-sectional in nature and had a small sample size of 50 participants [17]. A longitudinal study followed a group of 2230 adolescents, including nondrinkers, light drinkers, and heavy drinkers, for four years. Four basic executive functions were assessed between the ages of 11 and 19 years using laboratory-based neurocognitive tests at the start and after four years. No significant differences were found between drinking groups for any of the executive functions [18]. However, as the authors stated, these results do not rule out the possibility of deficits in neurocognitive functioning manifesting at some point later in life. 

To our knowledge, there has only been one study with a longitudinal design using a large group of young adults, which included all drinking ranges and various drinking patterns and focused on the association between alcohol consumption and cognitive performance [18]. The present study was designed to add to this knowledge. All data were collected through web-based and smartphone-based platforms. Cognitive performance was assessed using an online cognitive test battery. Ecological momentary assessment [19] allowed the study of drinking patterns, since drinking pattern may be important in determining the association between alcohol consumption and health outcomes [11,12,20]. We hypothesized that light to moderate drinkers obtain similar cognitive task scores as compared to abstainers, whereas heavy drinkers would obtain lower cognitive task scores.

## 2. Materials and Methods

Participants were students from Dutch Universities and Polytechnic Universities aged 18 to 24 years. Freshmen were excluded, because student life and drinking habits tend to be less stable during the first year at university [21]. Additional exclusion criteria, as assessed by a questionnaire at baseline, were alcohol/drug abuse or addiction as determined by being under treatment for substance related problems, under treatment by a psychiatrist for a current psychiatric disorder, and being pregnant or trying to get pregnant. A total of 1501 students were recruited, of whom 1193 were eligible. In order to receive reimbursement, participants were informed that they needed to complete all cognition tests and 60% of the triweekly alcohol consumption questions, which is a common compliance cut-off [22] and comprises in this study at least 7 out 12 responses during each four-week period. These criteria were met by 702 participants at Wave 1 and by 436 participants at Wave 2 (Figure 1).

### 2.1. Ethics

This study was executed in accordance with the Declaration of Helsinki. The study was evaluated by the Independent Ethics Committee (IEC) “Medisch Ethische Toetsings Commissie Brabant” on 25 August 2015 (number NW 2015-45) and considered not subject to the Medical Research Involving Human Subjects Act. A study-specific independent website was set up to recruit and inform potential participants to the study. After having read all information and after filling out the study specific informed consent form, participants were automatically referred to the baseline questionnaires to start the study. 

### 2.2. Alcohol Consumption and Drinking Pattern

Alcohol consumption was assessed in two ways. Firstly, by baseline questionnaire asking about quantity and frequency of drinking over the past year. Average alcohol consumption was calculated and expressed as grams of pure alcohol per day (for details, see Appendix A). Secondly, ecological momentary assessment (EMA) was used, which was employed through a smartphone application that participants downloaded on their phone. For four weeks, the EMA app sent three weekly notifications semirandomly, ensuring seven days of the week would be sampled and people would not receive two notifications on one day. The notification reminded participants to report the number of alcohol drinks consumed the previous day (day and evening, including the subsequent night). Notifications were sent at a random time between 10:00 and 20:00. Every glass of beer, wine, or mixed drink each counted as one Dutch standard drink, assuming that each contained 10 g of pure alcohol. The type of drink was not recorded. The weighted average pure alcohol consumption was calculated and used as a constant variable expressed as grams of alcohol per day. Average alcohol consumption based on the questionnaire was only used for comparison with the average alcohol consumption obtained through the EMA app. EMA app data were used for all other analyses.

In addition, EMA data were used to compute drinking patterns. First, drinking frequency was defined as the number of drinking occasions divided by the total number of EMA responses. Then, drinking intensity was defined as the percentage of drinking occasions at which women and men drank in binges. Binge drinking was defined as consuming more than 4 (women) or 6 (men) drinks on one occasion. Subsequently, for both frequency and intensity, median split was used as a cut-off. For frequency, this was 40% (i.e., in 40% of the EMA responses, participants indicated having had at least one drink the previous day). For intensity, this was also 40% (i.e., 40% of the drinking occasions were binge-drinking occasions). Finally, five groups were constructed:-Abstainers: participants reporting zero drinking occasions;-Occasional moderate drinkers: participants drinking on less than 40% of the occasions, of which less than 40% were binges;-Frequent moderate drinkers: participants drinking on more than 40% of the occasions, of which less than 40% were binges;-Occasional excessive drinkers: participants drinking on less than 40% of the occasions, of which more than 40% were binges;-Frequent excessive drinkers: participants drinking on more than 40% of the occasions, of which more than 40% were binges.

### 2.3. Cognition Tests

Cognition tests were presented through an online platform of Cambridge Brain Sciences (https://www.cambridgebrainsciences.com). These tests are specifically designed for remote cognition testing without the need of a laboratory visit, allowing efficient data collection on a large scale [23]. These tests have been shown to give a valid assessment of multiple cognitive domains [24,25]. Cognition tests consisted of the following tests: (1) ‘Odd one out task’, a deductive reasoning test based on a subset of problems from the Cattell Culture Fair Intelligence Test [26], (2) ‘Spatial span task’, a classical tool for measuring spatial short-term memory capacity based on the Corsi Block Tapping Task [27], (3) ‘Spatial rotation task’, a task often used for measuring the ability to manipulate objects spatially in mind testing concentration [28], (4) ‘Digit span’, a computerized variant on the verbal working memory component of the WAIS-R intelligence test testing working memory [29], (5) ‘Paired association task’, which is based on a paradigm that is commonly used to assess memory impairments in aging clinical populations [30], and (6) ‘Tree task’ or Spatial Planning Task, based on the Tower of London Task [31], which is widely used to measure executive function. A detailed description and visualization of each cognitive task is provided in Appendix A.

Outcomes were based on performance scores and not on reaction time as participants executed the tasks in an uncontrolled setting using different laptops or desktops and having different internet connections. Typically, each task started with an easy trial. If the participant answered correctly, the score and difficulty increased. If answered incorrectly, the difficulty decreased, and the score remained the same or decreased depending on the task. The task was ended after a set time or after three mistakes were made. Each task resulted in one final score.

### 2.4. Procedure

After inclusion, participants were asked to fill out the baseline questionnaires and instructed to download the EMA app. Data were subsequently collected in two study waves (Figure 2). In brief, Wave 1 consisted of 28 days of EMA and the subsequent cognition tests. 

Participants received an unique link via email through which they could access the online cognition tests. Cognition tests could be performed on a laptop or a computer, but not on a smartphone or tablet. For each wave, the cognitive testing session comprised two rounds. The first round of six tasks was to become acquainted with the tasks and avoid learning effects in the results. Then, a second round with the same six tasks was performed, and these data were used for analysis. Participants were instructed to make the tests alone, sober, and in a quiet environment. This sequence of EMA for 28 days followed by the online cognition tests was repeated one year later (Wave 2).

### 2.5. Laboratory Study

In addition to the large ‘at home’ study, a small laboratory study was conducted using the same cognition tests. Of the participants who had previously completed Wave 1 and Wave 2, a random sample of 41 participants, both polytechnic and university students, was invited to a laboratory at the Utrecht University Campus. Under more controlled conditions and with an experimenter present in an adjacent room, participants first performed the practice trials of the cognition tests, after which they completed the six cognition tests one last time. 

### 2.6. Data Analysis

Data analysis was carried out using R statistics (version 3.3.1). The study allowed for both cross-sectional as well as longitudinal analyses. Cross-sectional analysis was performed on the data obtained at Wave 1 (N = 702) and again on the data of Wave 2 (N = 436). Longitudinal analysis was performed on those that provided complete data during Wave 1 as well as during Wave 2 (N = 397). To determine which covariates to include, Pearson’s ρ (if normally distributed) or Spearman’s rank-order correlation (if ordinally distributed) were used to study correlations between covariates and cognitive task scores. Linear models (lm in R) were used to relate alcohol consumption to cognitive performance where also simultaneously two covariates were included—education level and gender and their pairwise interaction. Furthermore, the pairwise interactions of alcohol with the covariates were included as well, i.e., alcohol and sex, and alcohol and education level. Analyses of variances (ANOVA) were performed on these models to estimate the significance of the effects involved (i.e., alcohol and the covariates and their pairwise interactions). The ANOVA was two-sided with a significance level of 0.01 (*p*-value threshold for significance). 

To study the hypothesis that alcohol consumption affects performance in the cognition tasks, several ANOVAs were performed, as described above. In the first analysis, the effect of the mean alcohol intake during Wave 1 (obtained through EMA) on the outcome of each of the six cognition tasks in Wave 1 was investigated. In the second analysis, the effect of the mean alcohol intake during Wave 1 on the cognition task outcomes in Wave 2 was investigated. Similarly, the effect of the alcohol consumption data obtained during Wave 2 on the cognition task outcomes in Wave 2 was investigated. A similar analysis was performed on the effect of the change in alcohol consumption over a one-year period on the corresponding change in cognitive performance.

Post-hoc tests were performed to evaluate the variance within and between persons for each cognition test. Within-person variance was based on the scores obtained in Wave 1 and Wave 2, while between-person variance was based on the scores obtained in both waves combined. Similar analyses were performed on the data obtained in the laboratory study. As participants only visited the laboratory once, within-person variance was calculated using scores on the practice trial and on the test trial.

## 3. Results

### 3.1. Participants

A total of 702 participants completed the baseline questionnaires at Wave 1 and 436 participants completed Wave 2 (Table 1). Participants were on average 21 years of age. About 80% of the participants were Dutch Universities students, the remainder Polytechnic University students. 

### 3.2. Alcohol Consumption: Levels and Patterns

The percentage of participants over the five drinking patterns varied between 12 and 26% (Table 2). Percentage of university students was consistently high (66–85%) in all drinking patterns both during Wave 1 and Wave 2. 

Both during Wave 1 and Wave 2, there were fewer abstainers (10–12%) than alcohol consumers. Alcohol consumers mainly consisted of occasional excessive and frequent excessive drinkers (each about 25%) during Wave 1 and mainly consisted of occasional moderate and occasional excessive drinkers (23–29%) one year later. Although participants changed their drinking pattern, overall, there seemed to be no major shifts (for more than 50%) from one drinking pattern to another (Figure 3).

Alcohol consumption reported with ecological momentary assessment was higher compared to that reported in the baseline questionnaire. When drinking on a weekday (Monday through Thursday), average alcohol consumption as reported per baseline questionnaire was lower than alcohol consumption reported per EMA (31 g of alcohol versus 39 g of alcohol) (*p* < 0.01). When drinking on a weekend day (Friday through Sunday), this was 40 g of alcohol when reported in baseline questionnaires versus 45 g of alcohol when reported with EMA (*p* = 0.001). Given that EMA reports are made close in time to experience and therefore less subjective to bias, this suggests an underreporting in the baseline questionnaires of 11% and 21% for weekend days and weekdays, respectively. More statistical details are presented in Appendix A.

Alcohol consumption varied considerably over the drinking patterns, ranging between 0 g per day in the abstainer pattern up to 50 g alcohol per day in the frequent excessive pattern. 

### 3.3. Cognitive Performance

Cognition scores were within the ranges common for university students (Table 3) [32]. University students had higher scores than polytechnic school students in three out of six tests (*p* < 0.01, data not shown), with two tests showing no significant difference between the two education levels. There were no significant differences between Wave 1 and Wave 2 in the scores for the six cognition tests (Table 3). 

### 3.4. Alcohol Consumption and Cognitive Performance

Only for the spatial span task, and only for Wave 1, there was a borderline significant positive association between alcohol consumption and cognitive performance (*p* = 0.011). There were no other significant associations between average alcohol consumption and cognitive performance during both Wave 1 and Wave 2. Further, alcohol consumption during Wave 1 was not significantly associated with cognition scores in Wave 2 (see Appendix A for full statistical details). 

Using a linear mixed model, we analyzed the association between the change in average alcohol consumption (Wave 2 minus Wave 1) and the change in performance on each of the six cognition tasks (Wave 2 minus Wave 1). None of the associations was significant.

### 3.5. Drinking Patterns and Cognitive Performance

Similarly, there was no association between drinking pattern and cognition tests. A linear mixed model of all cognition tasks combined and alcohol-drinking pattern both during Wave 1 and Wave 2 did not show significant correlations (Appendix A). 

Further, changes in cognition task performance and changes in alcohol-drinking pattern were not significantly associated. 

### 3.6. Post Hoc Testing

Post-hoc tests of pairwise testing drinking categories per cognition test did not yield any significant differences. 

Variability on the cognitive tests was high, as indicated by high standard deviations. Post-hoc tests on the variance showed that the within-person variance component for the six tests was between 47% and 76% and was higher than the between-person variance component varying between 24% and 53%. In the laboratory study, the within-person variance component varied between 26% and 85% and was on average lower than the between-person variance component varying between 15% and 74%. 

Finally, we also assessed the association between alcohol consumption and cognitive performance in the laboratory. Similarly to the full study, no significant associations were found. 

## 4. Discussion

We hypothesized that light to moderate drinkers would obtain similar cognitive task scores as compared to abstainers, whereas heavy drinkers would obtain lower cognitive task scores. While the first part of our hypothesis was retained, we did not find lower cognitive task scores for heavy drinkers. In this study, we did not find any consistent association between alcohol consumption and cognitive performance in a large population-based sample of young Dutch adults. This observation was made both cross-sectionally as well as longitudinally after a one-year follow-up. These null findings were observed for both the average amount of alcohol consumed as well as for the various drinking patterns. However, the results of this study should be interpreted with caution, because the null findings of this study have to be viewed in light of the high variance of the cognition scores. 

The strengths of this study are the use of a large and homogeneous group of young adults: all students of similar age and similar level of education. This is relevant because cognitive performance largely depends on age and educational level. The group, however, spanned a large range of alcohol consumption and included various drinking patterns. Both the cross-sectional and longitudinal analysis used validated and well-recognized cognition tests. We selected these cognitive tests, since we considered them to provide a somewhat better indicator for day-to-day functioning and brain health as compared to functional MRI images showing changing patterns of blood circulation [14,15]. 

EMA may be a suitable methodology for alcohol consumption evaluation. EMA encompasses the brief but intensive repeated assessment of people’s thoughts, feelings, and behaviors in their real-world settings. The ecological validity of EMA data is considered high [19]. EMA reduces retrospective bias when assessing alcohol consumption, as suggested by higher consumptions as compared to consumptions recorded by regular questionnaire. EMA also has a low cognitive bias due to direct retrieval [33]. Furthermore, the repetitive data collection allowed the study of drinking patterns in addition to commonly reported average consumption levels. This is relevant since alcohol-drinking pattern may be an important determinant for the harmful effects of drinking, such as binge drinking [11,12,20]. 

Population surveys using questionnaires typically report underestimates of alcohol consumption of approximately 40–50%. Researchers adjust alcohol survey data to weight estimates such that they match alcohol sales or alcohol tax data. The current study suggests that underestimation of alcohol consumption in this population exists, but to a lesser extent than assumed in population surveys. EMA has been recognized as an alternative for assessing alcohol consumption in the natural environment [34]. 

Previous studies found inconsistent results on the relation between alcohol consumption and cognitive performance. The majority of studies indicate that long-term heavy drinking has strong negative associations with diseases of the brain such as dementia [35]. Many short-term studies indicate cognitive impairment in heavy binge drinkers as compared to nondrinking controls [8,9,10,11,12,13]. The outcome of comparing two groups differing in drinking habits, however, may depend on the selection criteria and may potentially be hampered by confounding. Excessive heavy drinking is usually accompanied by impulsive behaviors, risk-seeking behavior [36], and other traits [16] that may confound the association between alcohol consumption and cognitive performance. Some authors suggest that impaired cognitive performance may partly predict excessive alcohol consumption, whereas excessive alcohol consumption does not always predict impaired cognitive functioning [37]. 

Contrary to the differences in cognitive performance between heavy binge-drinkers and nonbinge-drinking controls, long-term moderate drinking has been associated with a reduced risk of dementia and a reduced risk of cognitive decline. Reviews of prospective studies showed that moderately drinking elderly have a decreased risk of dementia and cognitive decline [38,39]. Thus, after a very long follow-up, moderately drinking persons may be expected to show a less severe decline in cognitive performance as compared to those that drink excessively and abstainers. This suggests that there may be a J-shaped association between alcohol consumption and dementia and cognition, as has been described for cardiovascular diseases [40]. The risk reduction for dementia and age-related cognitive decline observed in the elderly may occur through a mechanism related to cardiovascular disease risk factors, whereas the cognitive impairment observed in young binge-drinking adults may occur through a mechanism related to neurotoxicity.

Our results correspond with those reported previously by Boelema et al. [18]. The null findings regarding the association between alcohol consumption and cognitive performance in that study were interpreted as being methodological in nature; the tests used may not have been sensitive enough to detect a potential cognitive performance reduction as a consequence of alcohol consumption. We also used conventional standard tests that are routinely used for cognitive performance evaluations. However, some aspects of our testing differed. Firstly, the tests were performed in an ‘at home situation’ as opposed to ‘at a testing facility’, which may have affected the results in various ways. For some individuals, performing cognitive tests in an environment that they are familiar with may positively influence performance. For others, the at home environment may have provided more distraction, or the lack of experimental control and the fact that no experimenter was present may have reduced focus and motivation, negatively affecting performance. All these factors may have affected test results and might explain the high within-person variability. Secondly, cognitive tests employed in the present study did not allow evaluation of aspects like reaction time, which may have contributed to a less complete test result. 

The cognition tests did seem to detect differences, since small significant differences were observed for education level. It is important, however, to extend these studies to enable detection of small differences in cognitive performance that may be induced by light and moderate alcohol drinking. Significant differences in cognition tests may be detected by decreasing the variability in the cognition test outcomes.

Although the study was set up with a group of students to obtain a high degree of homogeneity, this also has its limitations. The results obtained in this group cannot be generalized to the general population nor to specific other groups like persons with a low socioeconomic status. Specific groups may respond differently to alcohol consumption and may have more difficulty in adapting their drinking pattern whenever needed. In general, it has been extensively described that adolescents are less sensitive to the negative effects of alcohol, including cues that influence self-regulation of intake, but are more sensitive to positive effects, which may serve to reinforce or promote excessive intake [41]. This response to alcohol may promote the development of alcohol use disorders, a development university students may be less vulnerable to as compared to other groups of adolescents [7]. 

Our study design, however, had several limitations that warrant consideration. The null findings of this study have to be viewed in light of the high variance of the cognition scores. Whereas in the ‘real-life’ study, the within-person variability was higher than the between-person variability, in the laboratory study, the within-person variability was lower than the between-person variability. This suggests that the use of cognition tests in a ‘real-life’ setting may not have been suitable or sufficiently sensitive to detect a possible reduction in cognitive performance in association with alcohol consumption. Some of the tasks were, however, sensitive to education level, as university students outperformed polytechnic students, which would be expected as the former is a higher level of education. Furthermore, it is expected that the cognition tests used in this study might have been adequate to detect (possible) small differences in cognitive performance when used in a laboratory setting, provided a sufficiently large participants population.

In the present study, follow-up time was only one year. It would have been interesting to show in the same cohort that students who keep on drinking in a hazardous way will show cognitive impairment after many years. Boelema et al. [18], however, did report on cognitive performance after a four-year follow-up yet did not find indications for cognitive impairment in adolescent drinkers, including heavy drinkers. 

In conclusion, it is important to build on this study by reducing variance in online cognitive testing or by testing in a laboratory setting to better assess the association between light and moderate alcohol drinking and cognitive performance. In the present study, variance in cognitive performance was too large to detect an association, if any, between alcohol consumption and cognitive performance. Future studies should carefully consider both the context in which cognition is assessed as well as the type of tasks that are used.

## Figures and Tables

**Figure 1 nutrients-12-00200-f001:**
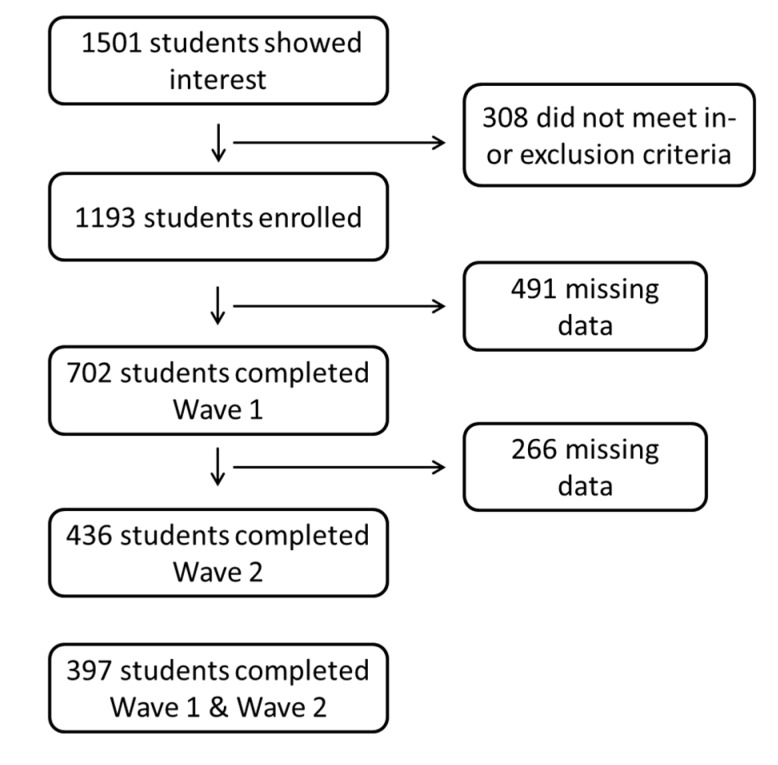
Student numbers throughout the study.

**Figure 2 nutrients-12-00200-f002:**
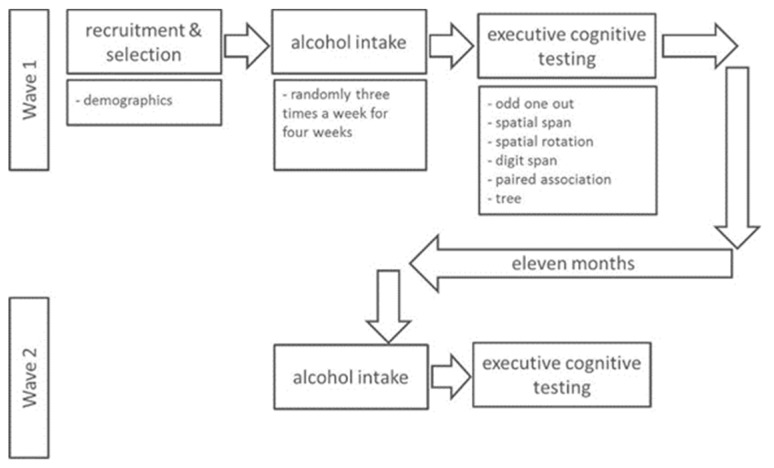
Study design.

**Figure 3 nutrients-12-00200-f003:**
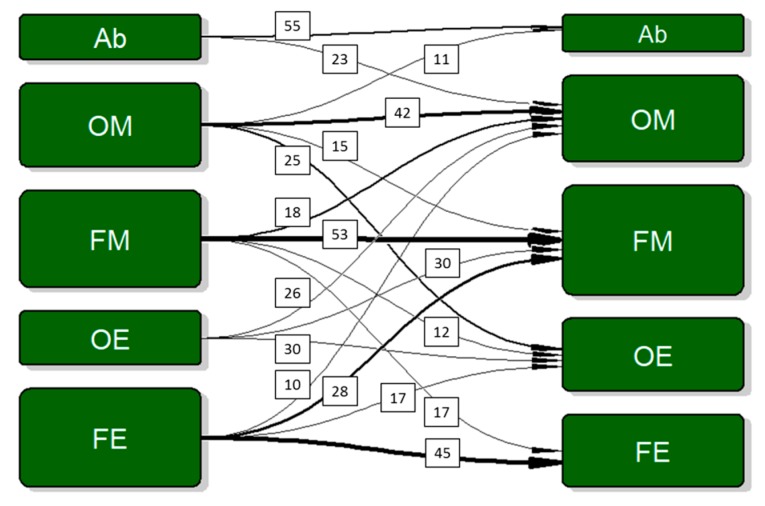
Participants’ drinking pattern at baseline (left column) and after a 1-year follow-up (right column). Width of the arrow and box size varies with but is not fully representative of the number of persons. Percentage of persons changing group is given for most but not all changes.

**Table 1 nutrients-12-00200-t001:** Population characteristics of participants in Wave 1 and Wave 2.

	Wave 1			Wave 2		
	Men	Women	Total	Men	Women	Total
Number	251	451	702	147	289	436
Age (y)	21.1 ± 1.8	20.8 ± 1.6	20.9 ± 1.6			
Education						
Polytechnic University (n)	44	120	164	21	70	91
University (n)	207	331	538	126	219	345

**Table 2 nutrients-12-00200-t002:** Number and percentage students, percentage females, percentage University students, and average alcohol consumption in Wave 1 and Wave 2 in the various drinking categories.

Characteristics	Abstainers	Occasional Moderate	Frequent Moderate	Occasional Excessive	Frequent Excessive
Number/% at Wave 1	84/12	159/23	108/15	168/24	183/26
Number/% at Wave 2	43/10	103/23	82/19	127/29	81/19
Female [%] at Wave 1	75	74	79	49	57
Female [%] at Wave 2	74	79	80	50	57
University student [%] at Wave 1	68	73	77	79	81
University student [%] at Wave 2	67	78	80	80	84
Alc. Cons. * at Wave 1 [gram/day]	0 ± 0	6 ± 6	15 ± 10	21 ± 15	50 ± 28
Alc. Cons. * at Wave 2 [gram/day]	0 ± 0	6 ± 5	16 ± 10	21 ± 13	48 ± 27

* Alc. Cons. = alcohol consumption.

**Table 3 nutrients-12-00200-t003:** Average scores at baseline and one-year follow-up for the six cognition tests.

Test	Wave 1	Wave 2	Δ (Wave 2) − (Wave 1) *	*p*-Value
	N = 702	N = 436	N = 398	
Digit span	6.7 ± 1.6	6.8 ± 1.9	0.1 ± 1.8	0.85
Odd one out	10.2 ± 3.9	10.1 ± 4.6	−0.1 ± 5.2	0.12
Paired association	5.2 ± 1.2	5.1 ± 1.4	−0.1 ± 1.6	0.62
Spatial rotation	105 ± 42	115 ± 46	12 ± 51	0.35
Spatial span	6.2 ± 1.2	6.1 ± 1.2	−0.0 ± 1.4	0.49
Tree task	29 ± 13	35 ± 13	5 ± 15	0.71

* None of the average scores in Wave 1 significantly differed from the average scores in Wave 2.

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
