# Peer review of "Alcohol Consumption, Drinking Patterns, and Cognitive Performance in Young Adults: A Cross-Sectional and Longitudinal Analysis"

_nutrients, 2020, doi:10.3390/nu12010200_

Round 1
Reviewer 1 Report
Review of manuscript:
ID: nutrients-674240; titled: Alcohol consumption, drinking patterns and cognitive performance in young adults: a cross-sectional and longitudinal analysis
General comment (originality, scientific accuracy, strengths and/or weaknesses):
I am pleased to review this interesting paper entitled: Alcohol consumption, drinking patterns and cognitive performance in young adults: a cross-sectional and longitudinal analysis. The manuscript is well written and has an original view of an important clinical problem, which are neurocognitive complications of alcohol drinking in large population based sample of young Dutch adults. However, authors did not find any statistically significant associations between alcohol consumption and cognitive performance, between change in alcohol consumption and cognitive performance in six tests after one year follow-up, as well as between drinking pattern and cognition tests outcomes. All of these discrepancies with tested hypothesis are well documented and discussed in detail. Authors enumerate potential limitations of results obtained and conclusions based on them, which were mainly short (one year) follow-up and online cognitive testing. As potential cause of null findings of the study authors proposed high variance of cognition scores, high capability of adaptation young adults to quantity of alcohol drunk and binge drinking pattern. Authors suggested also, that, on the one hand, initial low cognitive performance may be the cause of alcohol overuse, and, on the other hand, impairment in cognitive function secondary to even heavy alcohol consumption need many years.
Major corrections (main criticisms):
I suggest the repeat examination of study participants after ten years follow-up.
Minor revisions:
None
Author Response
We thank the reviewer or his/her positive scoring and positive review.
In response: We fully agree that it would be very interesting to perform a repeated evaluation after a longer period of time (e.g. after several years), but at the moment we lack the funding for such follow-up. Furthermore, all data has been anonymized and therefore there is no practical way to reconnect to the participants.
Reviewer 2 Report
The paper addressed an important topic: alcohol consumption and cognitive performance in young adults, especially since at the stage the brain is still maturing structurally and functionally.
It utilised both cross-sectional and longitudinal analysis to address the research question. Alcohol consumption was assessed in two different ways (triangulation) which increases reliability of data and a small laboratory study was also conducted.
Method, especially the analysis conducted was described in detail and results reported in a clear format. Thanks for providing the student number throughout the study and for clarifying the reason for missed data as “Compliance cut-off”. The integrity of the study will be improved if researchers provided more information regarding reimbursement. Was it monetary, credit for taking part etc
Results were laid out clearly in tables and figures. Well done.
The discussion was discussed in the light of previous and current findings. It was not exaggerated and conclusions drawn were appropriate. The authors also discussed the strength and limitations of the study.
Overall, a well written paper. Please see specific comments below for your attention.
Ethics:
Line 102. I will suggest that the authors name the specific website used to inform participants about the study rather than a blanket “A website was used to inform participants about the study and to obtain informed consent prior to the study start.”
I will suggest that Lines 369 to 371 be removed as it is not clear what it adds to the study. See below:
“Authors should discuss the results and how they can be interpreted in perspective of previous studies and of the working hypotheses. The findings and their implications should be discussed in the broadest context possible. Future research directions may also be highlighted.”
Author Response
We thank the reviewer for his/her positive scoring and positive review.
In response: The website used was a website specifically set up for this study (WWW.ABC-STUDIE.NL) at an external server and is not available anymore. The website was used for advertisement and recruitment and provided all study information, a digital informed consent form and all baseline questionnaires that were applied. After the students had given their consent form they were automatically referred to the baseline questionnaires. The baseline data has been exported to a protected TNO cloud environment and subsequently removed from that external server. Only anonymized data is stored in the protected cloud environment of TNO, where only the authors HH, AS, JK, and KH have access to.
Following up on the reviewer’s comment, we have changed the text to the following :
“A study-specific website was set up by an external IT consultant. The website was used to recruit for and inform potential participants about the study. After having read all information and after completing the study specific digital informed consent form, participants were automatically referred to the baseline questionnaires to start the study.”
Thank you for your comment on the lines 369 to 371. We have omitted these lines in the revised manuscript. These lines were not supposed to be there, but are remainders of the instructions to the authors in the manuscript outline provided by the publisher.
Reviewer 3 Report
This paper investigated longitudinally the association between alcohol consumption and cognitive function in young adult
Strengths include use of EMA to capture alcohol consumption and drinking patterns and longitudinal design.
There were two major comments
1) There is a lot of loss to follow up (43%) between wave 1 and wave 2 of the study. It is possible this is differential with respect to alcohol use and cognitive function. The authors should provide more information about differences in those who were followed up and those who were not with respect to baseline characteristics. The authors could also consider use of statistical techniques such as multiple imputation to assess the impact of missingness on their findings.
2) The authors do not provide the actual results from the analyses of alcohol use and cognitive function. They report in the text there were no significant associations (lines 243-257) but do not show the effect sizes or p values. Given investigating these associations is the aim of the paper then the actual results should be presented regardless of statistical significance.
Other comments:
3) Figure 1 - be good to include reasons why participants were excluded
4) Table 2 – The change in alcohol consumption between the groups was unclear – how could abstainers change by 4 grams per day. Is this amongst abstainers at baseline? The labelling of the table needs to be revised.
5) Line 171 – what was meant by representative selection - is this random sample? Or purposively sampled?
6) Table 3 – it would be good to include actual p values in this table
Author Response
We thank the reviewer for his/her valuable comments.
Comment 1:
We agree that a considerable number of volunteers were lost to follow-up. We do, however, not believe that this loss to follow-up is differential with respect to alcohol use and cognitive function. The main reason for that is the data presented in Table 2, 3 and Figure 3. In Table 2 it is shown that in wave 1 and wave 2 the percentage abstainers (12 vs 10%), occasional moderate (23 vs 23%), frequent moderate (15 vs 19%), occasional excessive (24 vs 29%) and frequent excessive (26 vs 19%) are very similar. Also, percentage female in the various drinking groups is consistent in the two waves (e.g., 57% wave 1 vs 57% wave 2 for frequent excessive drinkers), as is the percentage university students in each drinking group in the two waves (e.g., 81% wave 1 vs 84% wave 2 for frequent excessive drinkers). In addition, average alcohol consumption is also very similar for each drinking group in wave 1 as compared to wave 2. Table 3 shows there is no difference in cognition test score between wave 1 and wave 2. Also, figure 3 shows that the drinking pattern distribution does not change dramatically over the study year.
Because the percentage of missing data was relatively large we did not consider to apply this technique. Also as we did not observe a difference in cognition between drinking groups nor between wave 1 and wave 2, we were not triggered to consider such a technique. Moreover, imputation would potentially generate an artificial outcome, whereas we prefer to describe the observations made.
Comment 2:
We respectfully disagree with the reviewer that no actual results were given. Table 3 provides the actual outcomes of all 6 cognition tests for both wave 1 and wave 2. P-values were not provided in the submitted paper, but we agree that these p-values may be informative and therefore have added these to Table 3 in the revised manuscript.
Other comments:
Comment 3:
We have adapted Figure 1 to better describe why participants were excluded.
Comment 4:
We agree that the change in alcohol consumption is unclear and have therefore omitted this line from the table. We also revised the labelling of the table.
Comment 5:
Thank you for this comment, the sample was randomly chosen. This small group of students was compared to the main group in terms of their composition for gender and education. The composition of this small group was very similar to the composition of the large group in the main study. The text has been adapted to indicate that it was a random sample.
Comment 6:
As suggested P-values have been added to Table 3.
Reviewer 4 Report
Dear authors,
Your article is really interesting and well written. I really like your idea of considering an ecological methodology such as EMA to measure alcohol consumption and combining it with cognitive tasks. However, I have some major and minor concerns regarding your article which require a comprehensive revision of the article:
Major concerns:
The description of the methodology in its present form is insufficient. It is not transparent for the reader how alcohol consumption was recorded in the questionnaire and by EMA (drink specific or general frequency and quantity? pure alcohol/ethanol? Standard drink assessment or any other definition for ’one drink’ regarding to container size and alcohol content?). In addition, I have some concerns about your classification of drinking patterns. The WHO gives well-founded definitions of risky alcohol consumption behaviour that could be more appropriate than median splits (also in certain subpopulations such as students). Moreover, you should give a more detailed description of the cognitive tasks and best combine it with a visual presentation of the individual tasks in the appendix so that each reader is clear about how the tasks were structured and performed.
I am also confused by your statistical approach, which is not sufficiently described either. It is not clear which variables were used as dependent or independent variables (it is best to name them directly) and how variables were evaluated and what distribution they were subjected to (continuous, categorical, dichotomous). Also, you have not described what kind of ANOVA you conducted. I guess one problem is that you have calculated a large number of analyses, which in itself is very obscure. That's why I recommend that you reduce the analyses and focus on one or two important questions. That would be more than enough for one publication! Get to work in a more structured way: What is my main question and what are my side questions (all should be mentioned in the introduction)? What variables do I need to support these hypotheses? What analyses do I need to calculate to prove the hypotheses?
My last main concern is the presentation of your results. You often say that analyses were not significant without presenting the results in detail (F-value, p-value, confidence intervals, effect sizes if significant). It would be helpful to present the required test statistics for all analyses, i.e. significant and non-significant results, in tables. Otherwise, the reader will not be able to understand your approach or the results. Again, I think you're doing a great job, but you were overzealous. From my point of view, there are too many analyses for a single hypothesis. Focus again on exactly what your questions are and then answer them in your article. Then it can be a great contribution to alcohol research.
Minor concerns:
The abstract needs more specifications about what has been done and which population has been studied. Also missing is a conclusion relating to the main question of the work.
You write about a representative and a homogeneous sample, this is unfortunately not possible. Homogeneity can never be representative and population-based, it always represents only one specific subgroup.
Please state your hypotheses clearly: What do you want to investigate and what do you expect? In the current version of the article you have a hypothesis, but discuss many other aspects in your analyses - this is not understandable and obscure.
Why do you mention the alcohol questionnaire? This is not part of your hypothesis or introduction. You should drop it, even if it is really an interesting question. Source and R versions are missing.
How were 41 respondents (Labroratory Study) selected representative? It is not enough to say that the selection was "representative".
Did you consider sex (biological) or gender (socio-cultural)? You mentioned both....
Testing a "null hypothesis" is really critically due to a high beta error.
You may need to rethink the structure of your discussion. Start with a short summary, then discuss strengths and limitations, and then discuss your findings in relation to your hypotheses and the current literature. In the current version I miss the discussion of your results on your original hypothesis.
Young adults and older people should not be compared in terms of cognitive functions. So please do not discuss old people in the discussion on cognitive performance in young adults. These are two completely different populations.
I would recommend to have a look at recent WHO publications. You will find the most recent statistics on alcohol epidemiology for Europe: http://www.euro.who.int/en/health-topics/disease-prevention/alcohol-use/publications
Author Response
We thank the reviewer for his/her positive words on our manuscript and the methodology used.
Major concerns:
The reviewer considered our methodology description insufficient whereas other reviewers did not comment on this aspect of our manuscript or even indicated that the methodology was described in great detail. On alcohol consumption we have described the following: Average alcohol consumption was calculated and expressed as grams of pure alcohol per day. And: A glass of beer, wine, or mixed drink each counted as one drink, though the type of drink was not recorded. The weighted average of pure alcohol consumption was calculated and used as a constant variable expressed as grams of alcohol per day. Average alcohol consumption based on the questionnaire was only used for comparison with the average alcohol consumption obtained through the EMA app. EMA app data were used for all other analyses.
We did not indicate that alcohol consumption was calculated as pure alcohol. This has now been added to the text (see the word pure in bold in the above text).
We did not consider to use the WHO definitions of risky drinking, since we intended to study the effects of both quantity drunk and frequency of drinking, rather than risky drinking or not-risky drinking. If we would follow the reviewer’s suggestion, we would need a full re-structuring of the data and re-evaluation of all analyses, which we considered not feasible, especially within the timeframe provided. Further, we believed such a revision would be a too major change of the manuscript, which would be beyond what would be allowed in a revision round. Lastly, we do not believe such a re-structuring of the data would materially change our conclusions.
Regarding the cognitive tasks, we have added a detailed description and visual presentation in the appendix. We refer to the appendix in the method section.
In the analyses alcohol consumption was related to cognitive performance, alcohol consumption (continuous), gender (dichotomous) and education (dichotomous) were added as independent variables and cognitive performance as dependent variable. Regular ANOVA analysis (not ANCOVA nor a MANOVA) was performed to identify the significant variables in the model.
Several analyses were performed to confirm the results of the analyses related to the main research question. As the outcome was similar not all results of all the analyses are presented in the manuscript.
For clarity we have now included the p-values in table 3.
We believe we have clearly stated our main hypothesis at the end of the introduction section, namely: We hypothesized that light to moderate drinkers obtain similar cognitive task scores as compared to abstainers, whereas heavy drinkers would obtain lower cognitive task scores.
Minor concerns:
Abstract: In the abstract we have added more specifications regarding study procedures and population studied. Regarding the conclusion: our original formulation was chosen as we feel the main study question cannot be answered without referring to the high within person variance of the cognition scores. We have adapted the abstract to have a more clear conclusion regarding the main question pertaining the association between alcohol consumption and cognitive performance. However, we tried to maintain the note to the methodology to avoid erroneous interpretations of the main conclusion.
Representative and homogeneous: We have deleted the word ‘representative’ from the description of the laboratory study group and changed it into ‘random’. In the description of the main student group, we did not use the term ‘homogeneous’ as such, but used the term ‘relatively homogeneous’ to indicate that this was a ‘student-only’ population.
Hypothesis: see above
Alcohol questionnaire: We included the questionnaire to make sure we had enough variation in our population to test our hypothesis and would therefore prefer to keep it in the paper.
Source and R version are now provided: Publicly available statistical package ‘R’ version 3.3.1 and linear models using lm.
Laboratory study: the word ‘representative’ has been changed into ‘random’.
Gender/sex. We asked people to indicate whether they were a man or a woman. We did not explicitly ask for their biological nor their social-cultural sex, so we have used ‘gender’ throughout the manuscript.
Thank you for the suggestion to restructure the discussion. We believe that all elements listed by the reviewer were included in the discussion, although not necessarily in that order. We prefer to use the order provided.
Young and old people. The reviewer is right in saying that young and old people should not be compared in terms of cognition. We did, however, not intend to nor did compare young and old people in terms of cognition in our discussion. We discussed the effects of alcohol consumption on cognition in various groups of people. Interesting data are available on old people and therefore relevant to mention in the discussion section.
We have used the Dutch data on alcohol consumption because this study concerned Dutch students only. We had a look at the link provided and found many WHO publications. Looking through these publications, we realized that student alcohol consumption is highly variable in the European region (and in the world). Since it was not the intention to compare Dutch students with other European or World students we have not referred to those data.
Round 2
Reviewer 4 Report
Dear authors,
many thanks for your detailed response and successful revision of your article. I am pleased to see what great progress the article has made. From my view the manuscript is nearly ready to publish after taken into account some minor comments, please find below:
Alcohol assessment: I am still confused about your assessment of alcohol. You stated that each glass of beer/wine/mixed drinks was counted as one drink, but what was the container size and alcohol content of the drinks? And did the students all use the same definition? For example: One beer = 250 ml beer with 5% alc content; one wine = 100 ml with 12% alc content. I guess that the container sizes can vary greatly within alcoholic beverages (e.g., beer: 200 – 600 ml, light or strong beer). Please add the definition you used for clarification (you may have used the standard drink assessment, then please name it for the Netherlands). Please explain how alcohol consumption was recorded in the questionnaire (reference period, BSQF or GF assessment etc.); if applicable, please provide the questionnaire in the Appendix. Presentation of findings: Please provide test statistics for your analysis. Reporting only significance is not sufficient. The best way would be to present the results in the form of tables, possibly in an Appendix. Please add for all analysis: r/rho (correlations); F-statistics (ANOVAs) and beta’s (or equivalent coefficients for linear models). The presentation of test statistics is really important for the reader even if results are not significant and moreover, it contributes significantly to the transparency of your work. Discussion, first paragraph (p. 8): “While the first part of our hypothesis was confirmed, …”. The current state or the assumption to find no differences in the cognitive performance between light and moderate drinkers and abstainers is a null hypothesis. A null hypothesis never can be confirmed but only rejected or retained! This is one of the most important basic principles of inferential statistics (see also law of falsification and beta error). Please change this statement because this is a statistical bummer. Discussion, second paragraph (p. 8): Please state why it is a strength of your article that you included a homogenous group of young adults (e.g., cognitive performance depends on age and educational level). In general, the investigation of a homogenous group of respondents is rather a limitation, so a short explanation would be helpful for the reader. Coverage of alcohol in population-based surveys: Please state a source and specify your statement. It is true that population-based alcohol(!) surveys underestimate per capita consumption by 40-50%, however, at the aggregate level. I think I understand why you mention this aspect, as it is an important strength that distinguishes EMA from population-based surveys. However, I find this statement in its present form insufficient. It might be more appropriate to refer to other advantages of the EMA (e.g. low cognitive bias due to direct retrieval). For limitations in population-based alcohol surveys, for example: Greenfield & Kerr, 2008. Alcohol measurement methodology in epidemiology: recent advances and opportunities. Addiction, 103, 1082-1099. DOI: 1111/j.1360-0443.2008.02197.x Discussion, 5th paragraph, line 302: check spaces Alcohol and dementia in elderly: From my point of view, I still find it difficult to consider this population in the discussion, although I can understand your enthusiasm in the interesting findings. Conclusion: I guess the large variance in cognitive performance is one (important) contributor for hypothesis rejection, however, this will be not the only reason. Please rephrase the sentence as it currently sounds as if you were able to show that too large variance was the reason for non-significance (although we do not know if there really are no differences in cognitive performance in heavy drinkers…).Author Response
Please see the attachment
